# Effect of Steel Surface Roughness and Expanded Graphite Condition on Sliding Layer Formation

**DOI:** 10.3390/ma14112960

**Published:** 2021-05-30

**Authors:** Aleksandra Rewolińska, Karolina Perz, Grzegorz Kinal

**Affiliations:** Faculty of Civil and Transport Engineering, Poznan of University, PL-60965 Poznan, Poland; karolina.perz@put.poznan.pl (K.P.); grzegorz.kinal@put.poznan.pl (G.K.)

**Keywords:** expanded graphite, sliding layer, material transfer

## Abstract

The aim of the research was to evaluate the influence of the initial roughness of a steel pin cooperating with a graphite ring—dry and wet—on the mechanism of sliding layer formation. A ring–pin friction pair was used for the study, where the rings were made of expanded graphite, while the pins were made of acid-resistant steel. In the first case, the steel pin interacted with a dry graphite ring, and in the second case, the graphite rings were moist. To determine the effect of initial surface roughness, the pins were divided into three roughness groups. To determine changes in surface geometry due to material transfer, the Ra and Rz parameters were measured. This project investigated how the initial roughness value of the steel surface pin cooperating with expanded graphite influences the formation of the sliding layer. Increasing the initial roughness of the steel surface interacting with the graphite contributes to faster layer formation and reduced roughness. The state of the expanded graphite—dry and wet—influences the formation of the sliding layer of graphite—a wet graphite component causes a faster smoothing of the steel surface. The running time of the wear apparatus has an effect on the resulting layer. The highest roughness group is the most favorable from the viewpoint of sliding layer formation.

## 1. Introduction

The material transfer phenomenon in a graphite–metal pair consists of the transfer of the graphite to the metal surface, so that, with time, the pair can work in graphite–graphite configuration. The literature data on this phenomenon mainly refer to the material transfer of plastics containing graphite as filler [1,2]. The study shows that graphite-modified materials form a sliding layer on the metal surface, which is a factor retarding the destruction process on friction surfaces and increases the product value of pressures and speeds at which the exploitation of the tested materials is possible [3,4,5,6].

In contrast to graphite-modified materials, the phenomenon of transfer of pure graphite to a steel surface is not well recognized. The author [7] states that graphite, due to its properties, has the ability to adsorb on friction surfaces and form a strong sliding layer oriented in the direction of motion. However, the nature and course of the layer formation depend on many factors. These include, but are not limited to, the operating environment, the temperature, the material and the roughness of the mating surfaces [8,9]. An important factor affecting the resulting sliding layer is the moisture content of the sliding pair operating environment. Studies [10] performed for a friction pair, bronze and graphite–stainless steel composite, have shown that the presence of water significantly reduces surface wear. The positive effect of water on the graphite–graphite pair work was confirmed by another study [11]. An example of sliding pairs widely used in aqueous environments are face seals operating in a ceramic–graphite ring configuration [12,13]. According to one theory [7] graphite shows good lubricating properties due to water adsorption. The adsorbed water molecules on the main sliding planes reduce the adhesion between them and thus reduce the coefficient of friction as they slide [11,14,15]. The problem turns out to be that too much water washes away the resulting graphite wear particles, which are necessary for the formation of the sliding layer [16]. The existence of the problem is confirmed by studies [17,18], which indicate that a small amount of water in the sliding pair contributes to a decrease in friction, while too much contributes to an increase in wear. Temperature also has an influence on the layer formation in the pair containing graphite material [19]. For selected material pairs, it was found that the higher the temperature, the higher the coefficient of friction [20]. As the temperature increases, the contact area between the mating materials increases due to the destruction of the protective sliding layer. Moreover, graphite oxidizes at high temperatures, causing instability of the resulting layer. The authors of [21] showed that at temperatures of up to 50 °C, the formed layer is visible as a slight tarnishing of the metal surface. Above 100 °C the resulting layer is thicker, while above 300 °C the graphite covers the original machining marks. The reason for this is the increase in graphite consumption at higher temperatures. Research [20] confirms that the wear rate of the graphite surface increases with increasing temperature. The resulting wear products are embedded in surface unevenness and form a film. In addition to the effect of temperature on the resulting layer, the authors [22,23] also investigated the effect of the roughness of the surface mating with the graphite component. Observing steel mandrels operating at 150 °C in valve stuffing boxes in association with rings made of expanded graphite, it was noted that the surface roughness of the mandrels affects the layer appearance. The polished chrome surfaces with a roughness of Ra = 0.15 μm had virtually no traces of the graphite layer. The material did not have the ability to anchor into the microcavities. As the mandrel roughness increased to Ra = 2.6 μm, the visibility and thickness of the layer had also increased. However, at ambient temperature, it was difficult to find a correlation between the layer formed and the surface roughness of the mandrel. The effect of surface roughness on the resulting layer was also addressed in [24]. Surfaces with higher roughness parameters were judged to be more favorable because they acted as microscopic reservoirs of solid lubricants. The thick and heterogeneous layers formed remained intact and withstood higher shear loads. The study showed that even though the adhesion was poor, the ability to anchor the graphite particles in the microcavities contributed to improved layer durability. Subsequent researchers [25] evaluated the effect of roughness and time of operation on the layer formed. They observed that in the early stages of the operation of carbon materials with steel, the wear rate of graphite increased. In addition, the process was characterized by a large number of fine carbon particles that were outside the contact zone. In further operation of materials, the carbon impurities formed a layer of compacted particles on both the carbon surface and the opposite harder mating surface. The presence of the transfer layer provided a cushioning effect and a decrease in the wear coefficient. In the study of [26], the author observed that a significant amount of graphite wear particles produced remain in the contact zone. The wear particles, after repeated deformation and fragmentation, agglomerated at appropriate locations on the wear surfaces and formed a stable layer. The sudden increase in the coefficient of friction was due to the disruption of the graphite layer. An interesting conclusion regarding roughness was made by the authors of [16] who reported that if the difference in hardness is greater than 20%, the surface roughness will play an important role in reducing wear on mating surfaces.

Summarizing the analysis of the state of the art, it can be said that the literature quite unambiguously indicates a positive effect of the formed graphite layer on the durability of the cooperating materials. The effect of the layer depends on a number of factors, including the roughness of the mating surfaces, the temperature and the operating environment. The large number of physical and chemical factors affecting graphite transfer means that a comprehensive explanation of this phenomenon is lacking to date. It is also difficult to determine which of the elementary wear phenomena dominates under specific friction conditions.

This paper presents the results of a preliminary study of the material transfer phenomenon in the pair of expanded graphite (dry and wet) and steel with different roughness to determine the mechanism of graphite sliding layer formation on a steel component.

## 2. Materials and Methods

A ring–pin pair was used for the study (Figure 1). The rings were made of expanded graphite, which is the carbon material most commonly used in industry due to its easy and inexpensive manufacturing process [27,28,29,30,31]. Unlike traditional graphite, expanded graphite is a material with a lower degree of crystalline ordering; it is porous, with a low bulk density of about 0.001 g/cm^3,^ and susceptible to rolling. Expanded graphite can increase in volume by hundreds or even thousands of times its original dimensions. For example, graphite flakes with thicknesses in the range of 0.4–60 µm in their initial state can increase their dimensions up to about 20 mm [27].

The pins were made of acid-resistant steel—AISI 304. Information about the tested materials is presented in Table 1.

The tests were performed on a pin–ring test stand (Figure 1), where the pin moved in reciprocating motion on the surface of the ring made of expanded graphite. The reciprocating motion of the pin was achieved using an eccentric mechanism.

The tests were conducted with a load of 18.5 N and with an average pin velocity of 25 mm/s over a 6 mm section. The friction surfaces of the pins were tested before mating, after static load at rest for 30 s, and then after 10, 30 s and after 5, 15 and 30 min of mating. In the first stage, friction was dry (i.e., the steel pin cooperated with the dry graphite ring), and in the second stage, the graphite rings were moist (they were soaked in water for 24 h before the test). Before the test, the rings were removed from the container with water and dried on blotting paper. The working conditions are shown in Table 2. The designed working conditions of the graphite rings and steel pins correspond to the actual conditions of their work. This association works for the sealing of industrial fittings. The seals used in the valves work in a motion–rest system. This system may be disadvantageous, e.g., when it is the cause of vibrations causing disturbances in the movement of rubbing elements, the so-called the stick – slip phenomenon, which is characteristic of graphite.

In order to determine the influence of the initial roughness of the pin on the mechanism of formation of the sliding layer, tests were carried out in which the graphite ring had the same roughness parameters in each case when cooperating with steel pins, the surface of which was made in three different roughness groups, hereinafter designated I, II, III. To determine the changes in surface geometry due to material transfer, the following pin surface roughness parameters were measured: Ra and Rz (Table 3).

Three repetitions of roughness measurements were performed for each surface. After the roughness measurements, the steel surfaces of the samples were subjected to observational studies using a scanning microscope (Nikon, Tokyo, Japan). This enabled a preliminary assessment of the formed sliding layer. Measurements of 2D surface stereometry were made with a ZAISS contact profilometer (Carl Zeiss AG, Oberkochen, Germany) equipped with heads with an induction transducer and SUFORM software by SAJD METROLOGIA (Kielce, Poland), which allows for measurements and analysis of deviations of straightness and surface roughness. The measurements were made by determining the sampling length λc = 0.8 mm, where the measuring length Ln was adequately 4 mm.

## 3. Results and Discussion

### 3.1. Surface Roughness

The surface roughness parameters of steel pins before and after testing for dry and wet samples are shown in Figure 2 and Figure 3. The results are for pins of roughness group I.

Under dry running conditions, a significant increase in the surface roughness of the pin occurs after 5 min. In further operation, the roughness does not drop below the initial value. The surface of a steel sample working with a moist ring shows a decrease in roughness after 15 min. At this stage of research the theory already confirms that the presence of water affects the lubricating properties of these materials [11,14,15].

A summary of the roughness parameters Ra and Rz of the mandrels working in dry conditions and in cooperation with moist rings for roughness group I is shown in Figure 4 and Figure 5.

The surface tests performed for the roughness group I showed that dry operation increased the roughness of the steel samples. When the steel pins were worked with moist counter samples, the surface roughness began to decrease after 5 min of operation. The results of Ra and Rz parameters for group II roughness are shown in Figure 6 and Figure 7.

During dry operation, the highest increase in roughness occured after 5 min. After 30 min, a decrease in roughness values was observed compared to group I. The operation of the friction node with moist ring showed a significant decrease in roughness after 30 min of operation. The summary of the roughness parameters of the pins working in dry and in association with moist rings are shown in Figure 8 and Figure 9.

The greatest differences are seen after 5 min of friction node operation. The longer running time resulted in a decrease in the surface roughness of the pin. After 30 min of operation, the friction node with moist rings had a lower roughness than before the test. A decrease in roughness was also noted for the friction node running dry. The performed pin surface roughness measurements for group III are shown in Figure 10 and Figure 11.

During dry operation, a decrease in baseline roughness was noted after 30 min of operation. This distinguishes group III from the previous groups, which had higher roughness values. For samples mated with moist rings, a decrease in roughness was noted after only 5 min of operation. A summary of the roughness parameters of pins operating in dry and with moist rings for group III is shown in Figure 12 and Figure 13.

Group III had the highest surface roughness. There was no significant increase in roughness for the dry worked samples. However, the roughness decreased after 30 min of operation. Samples working with moist rings had a lower roughness than the initial roughness after only 5 min of operation. This condition persisted until the end of the friction test. The roughness of dry and wet pins was comparable after 30 min of operation.

In all the analyzed groups of roughness, the working time of the graphite–steel pair had an impact on the sliding layer formed, which was noted in investigation [25].

In order to evaluate the effect of the initial roughness and the operating conditions (dry and moist counter samples) on the sliding layer formation, the results of the surface roughness measurements of the tested pins after 30 min of association operation were summarized (Figure 14 and Figure 15).

The graph (Figure 14) shows that after 30 min of operation, the initial roughness of group I increased, while for group II it reached the initial value. Group III had a decrease in roughness from the baseline value.

The steel surfaces interacting with the moist counter samples after 30 min of operation for all groups had lower roughness compared to the initial roughness.

### 3.2. Physical Model of Sliding Layer Formation

The analysis of the chemical composition of the layer carried out in the first stage showed that the material from which the layer came into being was coal. Roughness profilograms and photographs of the pin surfaces after testing allow us to present a preliminary model of the formation of a graphite sliding layer due to the transfer of graphite to the steel component. In the initial stage of the research, graphite particles are anchored to the uppermost tips of the microcavities (1). Then, as a result of the movement, the amount of graphite material on the steel surface increases (2). After 5 min of work, the visible grooves of the indoor surface are filled with graphite. However, the steel vertices are still noticeable (3). This phenomenon is practically imperceptible after 30 min of work of samples cooperating with wet rings (4). The photos, along with the diagrams, are summarized in Table 4.

After 30 s of static contact, there is a transfer of graphite particles to the metal surfaces for both dry ring contact and moist ring contact. However, it is difficult to see any regularity in the arrangement of the graphite particles. An initial movement lasting 10 s brings no significant change. A movement lasting 60 s causes the roughness of the pin to increase. The reason for this change is an increase in the amount of graphite material transferred systematically to the steel surface—specifically, first of all on the tips of the microcavities. The highest values of the roughness parameters are obtained after five minutes of movement. After this time, the profile of the surface roughness outline of the pin begins to smooth out. This phenomenon occurs much better with samples working with moist rings. After 15 min of operation, these samples had a lower surface roughness than before the test.

Studies [22,23] suggested that the resulting sliding layer is visible faster for higher surface roughness. This phenomenon is confirmed by the obtained research results.

In conclusion, it can be said that the graphite layer is formed by three mechanisms. The first one is the adhesive transfer of the expanded graphite material to the uppermost tips of the microcavities. In addition to this, a second process occurs at the same time, which is the formation of a sliding layer of loose, fragmented particles, which, as a result of their separation from the surface of graphite rings, accumulate in the recesses of the steel sample microcavities.

The third mechanism occurs when large graphite particles are detached from the ring surface due to the cutting action of the tips of the steel surface microcavities and placed in the recesses between the tips of the microcavities. This proves that in addition to the adhesive interactions between graphite and steel substrate, mechanical interactions play an important role in the formation of the sliding layer, as a result of which the particles detached from the ring surface and not removed from the friction surface participate in the sliding layer formation [24]. During subsequent movements, these particles are pressed between the tips and smeared on their surface. Further loose particles can also adhere to them by means of adhesion. The layer formed in this way consists of long strands at the tips of the microcavities and small and larger particles in the recesses. During further movements, the loose graphite particles are pressed against the graphite particles already in the recesses.

During the operation of steel samples with both dry and moist rings, wear products are formed, which are carried out of the area friction node. Loose particles of these products are visible to the unaided eye on the surface of the rings. A significant number of particles outside the contact zone were mentioned in works [25,26].

## 4. Conclusions

The results obtained allow us to formulate the following conclusions:1The initial roughness value of the steel surface working with expanded graphite affects the formation of the sliding layer and thus the roughness of the steel surface after testing. Increasing the initial roughness of the steel surface interacting with the graphite contributes to faster layer formation and reduced roughness. This is seen especially for the pins working with dry graphite.2The state of the expanded graphite (dry or wet) influences the formation of the sliding layer of graphite—a wet graphite component causes a faster smoothing of the steel surface.3The running time of the friction node has an effect on the resulting layer:
-The transfer of material to the metal surface already occurs at static contact;-A significant increase in the surface roughness occurs after 5 min operation, especially for dry expanded graphite;-After 30 min of operation, samples interacting with moist expanded graphite have a lower roughness than before the test.4Comparing the roughness groups selected for the study, it can be indicated that group III is the most favorable from the point of view of the formation of the sliding layer. In the case of this group, the roughness of the pins of the dry samples after 30 min of work dropped below the initial roughness. This phenomenon was not observed for groups I and II. On the other hand, the greatest decrease in the pin roughness occurred for the moist samples.

## Figures and Tables

**Figure 1 materials-14-02960-f001:**
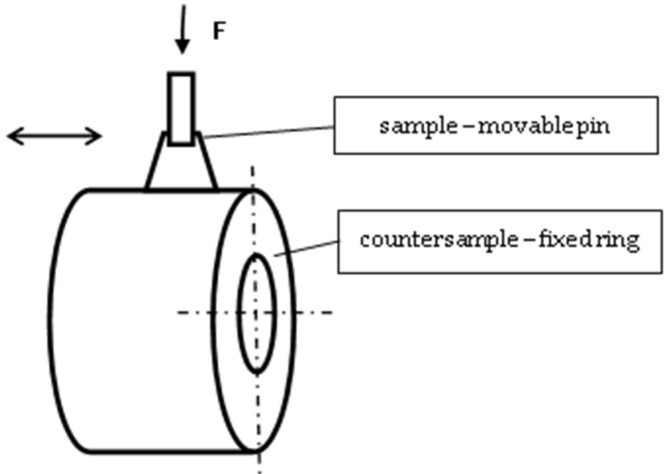
Scheme of analyzed friction node.

**Figure 2 materials-14-02960-f002:**
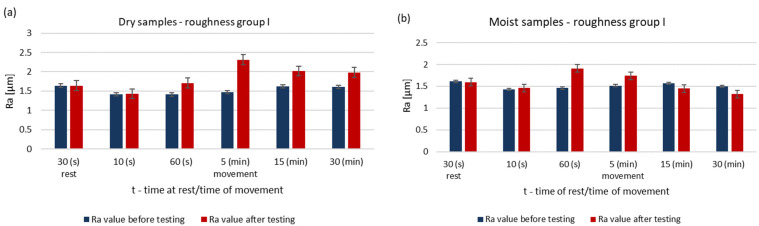
Changes in the value of the parameter Ra of steel pins for the roughness group I (Ra = 1.41–1.64 μm): (**a**) tests performed in dry conditions, (**b**) tests performed with moist graphite rings.

**Figure 3 materials-14-02960-f003:**
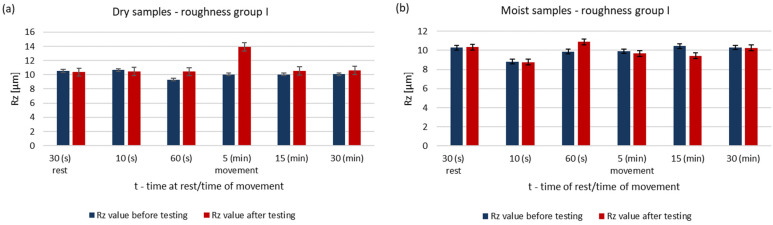
Changes in the value of the parameter Rz of steel pins for the roughness group I (Rz = 9.32–10.89 μm): (**a**) tests performed in dry conditions, (**b**) tests performed with moist graphite rings.

**Figure 4 materials-14-02960-f004:**
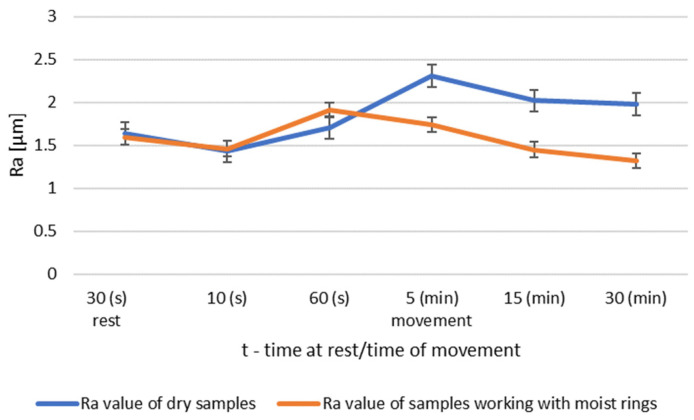
Comparison of Ra values of dry and wet ring samples for roughness group I.

**Figure 5 materials-14-02960-f005:**
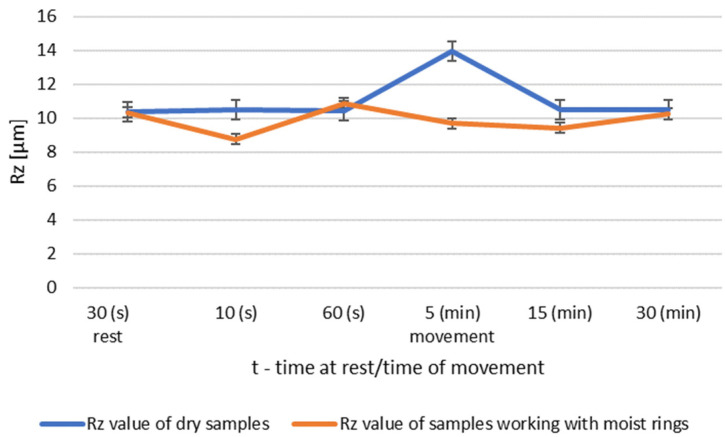
Comparison of Rz values of dry and wet ring samples for roughness group I.

**Figure 6 materials-14-02960-f006:**
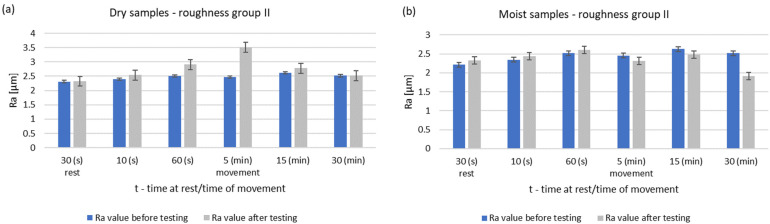
Changes in the value of the parameter Ra of steel pins for roughness group II (Ra = 2.31–2.62 μm): (**a**) tests performed in dry conditions, (**b**) tests performed with moist graphite rings.

**Figure 7 materials-14-02960-f007:**
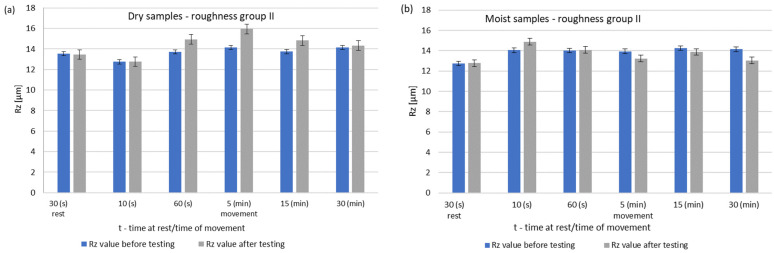
Changes in the value of the parameter Rz of steel pins for roughness group II (Ra = 2.75–14.15 μm): (**a**) tests performed in dry conditions, (**b**) tests performed with moist graphite rings.

**Figure 8 materials-14-02960-f008:**
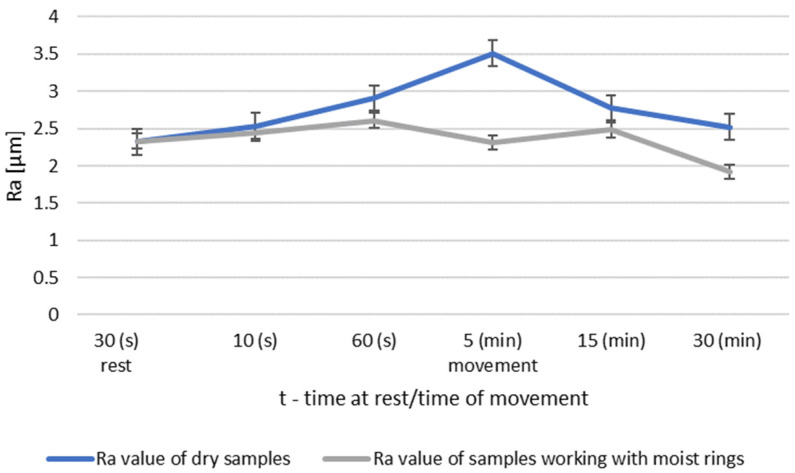
Comparison of Ra values of dry and wet ring samples for group II roughness.

**Figure 9 materials-14-02960-f009:**
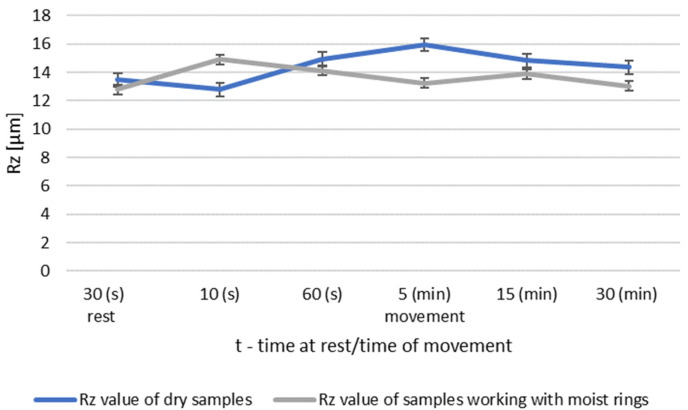
Comparison of Rz values of dry and wet ring samples for group II roughness.

**Figure 10 materials-14-02960-f010:**
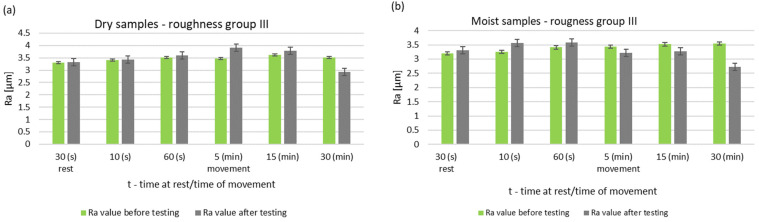
Changes in the value of the parameter Ra of steel pins for roughness group III (Ra = 3.21–3.62 μm): (**a**) tests performed in dry conditions, (**b**) tests performed with moist graphite rings.

**Figure 11 materials-14-02960-f011:**
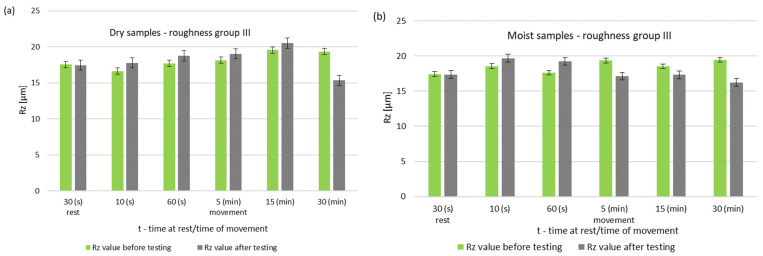
Changes in the value of the parameter Rz of steel pins for roughness group III (Rz = 17.54–19.55 μm): (**a**) tests performed in dry conditions, (**b**) tests performed with moist graphite rings.

**Figure 12 materials-14-02960-f012:**
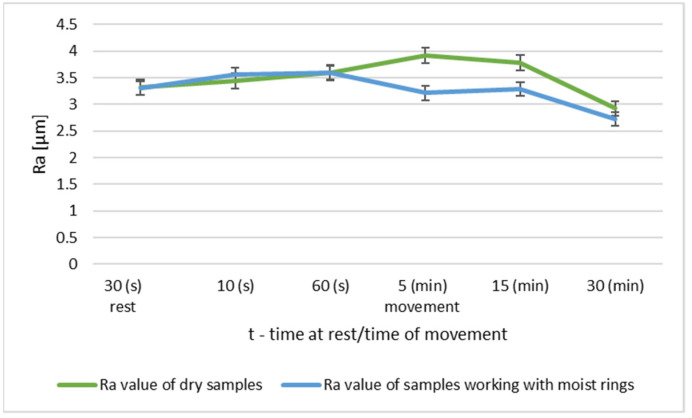
Comparison of Ra values of dry and wet ring samples for group III roughness.

**Figure 13 materials-14-02960-f013:**
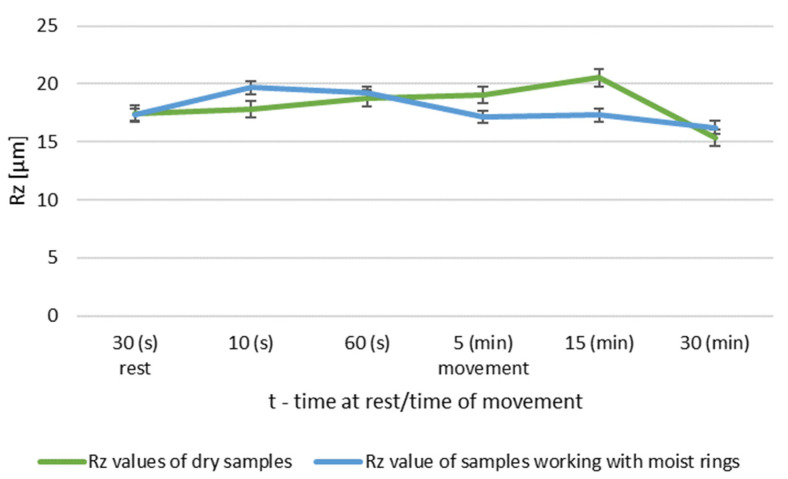
Comparison of Rz values of dry and wet ring samples for group III roughness.

**Figure 14 materials-14-02960-f014:**
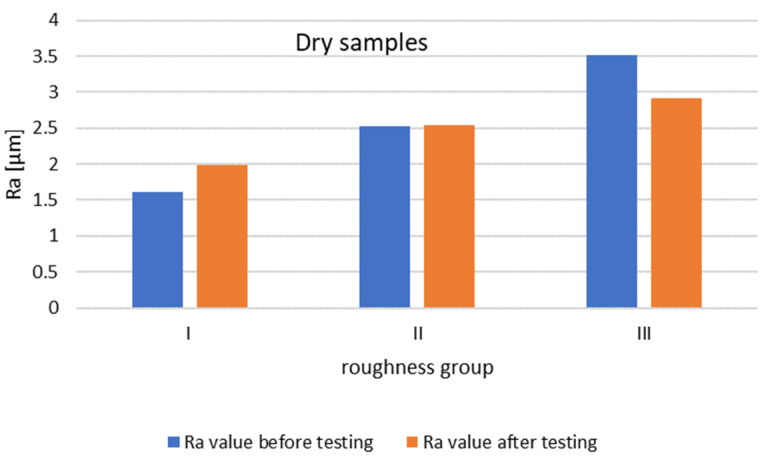
Change in Ra value of the pin surface after 30 min of dry running.

**Figure 15 materials-14-02960-f015:**
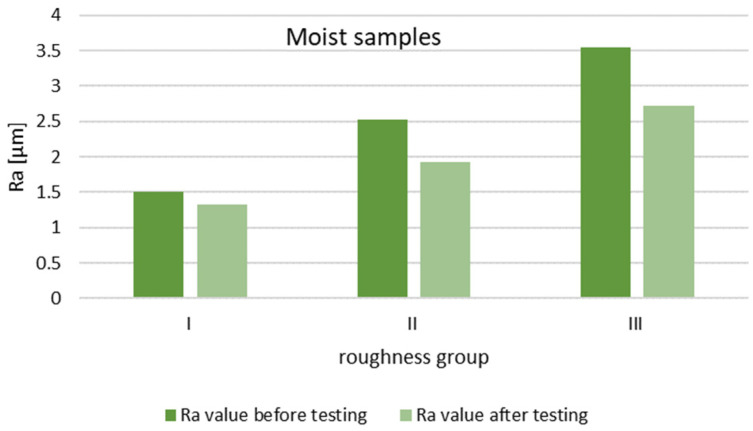
Change in Ra values of pin surface parameters after 30 min of operation with moist rings.

**Table 1 materials-14-02960-t001:** Material data sample and counter sample.

	Material	Shape	Dimensions (mm)	Hardness	Density (g/cm^3^)
Sample	acid resistant steel	countersunk pin	ϕ10 × 30	29–30 HRC	7.8
Counter Sample	expandedgraphite	ring	ϕ60 × 40 × 10	20 HV0,05	1.6

**Table 2 materials-14-02960-t002:** Working conditions of friction node during the tests.

Working Conditions	Rest	Movement				
	(s)	(s)	(s)	(min)	(min)	(min)
Dry Sample	30	10	30	5	15	30
Moist Sample						

**Table 3 materials-14-02960-t003:** Values of steel pin output roughness parameters.

Parameter/Roughness	Roughness Group		
	I	II	III
R_a_[μm]	1.41–1.64	2.31–2.62	3.21–3.62
R_z_[μm]	9.32–10.89	12.75–14.15	17.54–19.55

**Table 4 materials-14-02960-t004:** Schematic diagram of the formation of a graphite sliding layer on the surface of a steel sample and photographs of its surface after interaction with a dry and moist counter sample of expanded graphite (roughness group I).

Conditions of Cooperation				
Rest/Movement Duration	Samples operating in dry conditions		Samples operating with moist rings	
Rest 30 s (1)	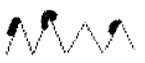	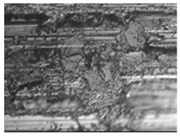	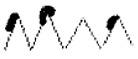	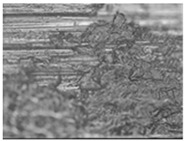
Movement 10–60 s (2)	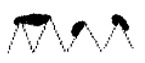	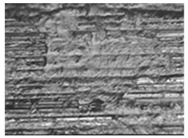	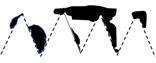	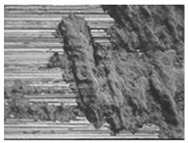
Movement 5 min. (3)	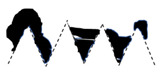	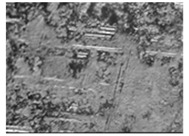	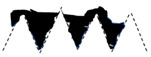	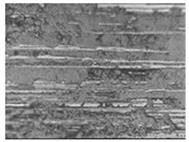
Movement 15–30 min. (4)	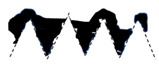	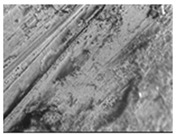	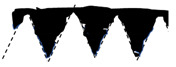	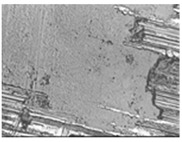

## Data Availability

Not applicable.

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
