# Peer review of "Effect of Steel Surface Roughness and Expanded Graphite Condition on Sliding Layer Formation"

_materials, 2021, doi:10.3390/ma14112960_

Round 1

Reviewer 1 Report

You have done an interesting research, but the scheme design, experimental process and result analysis of the article still need to be improved, such as below.

  1. What are the reasons for choosing expanded graphite as samples? Is because it has been widely used or has unique advantages?
  2. Is there any basis for the design of test parameters in Table 2? Does it correspond to the actual application?
  3. This paper only analyzes the physical formation of the sliding layer, whether there will be differences in chemical composition under different conditions? It is suggested to characterize the chemical state of the layer.
  4. There are many problems in the language of the article, such as too long sentences, which need to be carefully checked and revised by the author.

Author Response

Good morning,

Thank You very much for Your review. I have included the answers in the file.

Best regards.

Ola RewoliÅ„ska 

Reviewer 2 Report

This work has investigated the influence of initial surface roughness and graphite conditions on the sliding layer formation by comparing the surface roughness change after the Pin-on-disk test. The results are well explained. The major comments are listed below.

  1. Some of the experimental conditions are not well introduced. What is the composition of the “acid-resistant steel” used in this study? How was the Ra, Rz measured, optical or contact method, measuring area? How many samples are included in each roughness group?
  2. Why surface roughness change after the pin-on-disk test reflects the sliding layer formation? The logical is not clear since material transfer due to sliding contact does not always guarantee a better surface finish.
  3. The theory proposed in Table. 3 is not well supported or related to the experimental results.
  4. The author may want to cite some more recent literature review to give a better background of the graphite tribological study.

Author Response

Good morning,

Thank You very much for Your review. I have included the answers in the file.

Best regards,

Ola RewoliÅ„ska 

Reviewer 3 Report

Please find the attached file for detailed comments

Author Response

Good morning,

Thank You very much for Your review. I have included the answers in the file (an attachment).

Best regards 

Ola Rewolińska

Round 2

Reviewer 1 Report

The author has explained the problems existing in the article, introduced the scheme design.

In addition, the text should be carefully checked and polished.